# Photophysiological and Oxidative Responses of the Symbiotic Estuarine Anemone *Anthopleura hermaphroditica* to the Impact of UV Radiation and Salinity: Field and Laboratory Approaches

**DOI:** 10.3390/antiox13101239

**Published:** 2024-10-15

**Authors:** Edgardo Cruces, Víctor M. Cubillos, Eduardo Ramírez-Kushel, Jaime A. Montory, Daniela A. Mardones, Oscar R. Chaparro, Francisco J. Paredes, Ignacio Echeverría-Pérez, Luis P. Salas-Yanquin, Joseline A. Büchner-Miranda

**Affiliations:** 1Centro de Investigaciones Costeras, Universidad de Atacama (CIC-UDA), Avenida Copayapu 485, Copiapó 1530000, Chile; 2Instituto de Ciencias Marinas y Limnológicas, Facultad de Ciencias, Universidad Austral de Chile, Valdivia 5090000, Chile; 3Laboratorio Costero de Recursos Acuáticos de Calfuco, Facultad de Ciencias, Universidad Austral de Chile, Valdivia 5090000, Chile; 4Centro I~mar, Universidad de Los Lagos, Casilla 557, Puerto Montt 5480000, Chile

**Keywords:** symbiotic estuarine anemone, UV radiation, salinity, *Anthopleura hermaphroditica*, photosynthesis, antioxidant metabolism, oxidative damage

## Abstract

The estuarine anemone *Anthopleura hermaphroditica* and its symbiont *Philozoon anthopleurum* are continuously exposed to intense fluctuations in solar radiation and salinity owing to tidal changes. The aim of this study was to evaluate the effects of the tidal cycle, solar radiation, and salinity fluctuations on the photosynthetic and cellular responses (lipid peroxidation, total phenolic compounds, and antioxidant activity) of the symbiont complex over a 24 h period in the Quempillén River Estuary. Additionally, laboratory experiments were conducted to determine the specific photobiological responses to photosynthetically active radiation (PAR), ultraviolet radiation (UVR), and salinity. Our field results showed that the photosynthetic parameters of the symbiont complex decreased with increasing ambient radiation; however, no relationship was observed with changes in salinity. Increased peroxidative damage, total phenolic compound levels, and antioxidant activity were mainly related to increased UVR and, to a lesser extent, PAR. During the dark period, only PAR-exposed organisms returned to the basal levels of photosynthesis and cell damage. Laboratory exposure confirmed the deleterious effects of UVR on the photosynthetic response. The present study suggests that the ability of *A. hermaphroditica* to acclimate to natural radiation stress is mediated by the concerted action of various physiological mechanisms that occur at different times of the day, under varying levels of environmental stress.

## 1. Introduction

Variability in environmental fluctuations in estuarine ecosystems imposes a series of physiological stressors on sessile organisms. Similarly, diurnal tidal changes in these ecosystems can modify the availability of light, salinity, and temperature of the water column over a short period of time. In the Quempillén Estuary (41°52′ S; 73°43′ W/Ancud-Chiloé Island, southern Chile), tidal changes can expose organisms not only to fluctuating salinities (between 7 and 34 Practical Salinity Units (PSU)) [1,2], but also to high ambient radiation [3]. Sessile organisms in estuarine zones have evolved a series of physiological mechanisms that allow them to rapidly adjust their metabolism to survive changes in environmental conditions [2,4]. The Quempillén Estuary is a dynamic and fluctuating system that offers a unique environment for studying the influence of these factors on the photosynthetic and cellular responses of anthozoans. A inhabitant of the estuarine zones of southern Chile is the anemone *Anthopleura hermaphroditica*, a small anthozoan (1–2 cm) with an endosymbiotic relationship with the dinoflagellate *Philozoon anthopleurum* [5,6,7]. Anemones are highly vulnerable to environmental fluctuations because of their sessile nature and their soft tissues are directly exposed to environmental stress [8]. During low tide, anemones can be simultaneously exposed to low salinity and high levels of UVR, abiotic factors that induce the accumulation of reactive oxygen species (ROS) in the cell, causing structural and functional damage to macromolecules (e.g., lipids, proteins, and nucleic acids) [9]. Understanding the mechanisms underlying adaptation to light and salinity stress in the estuarine symbiont *A. hermaphroditica* will contribute to broader ecological and conservation efforts.

In response to solar radiation and salinity stress, *A. hermaphroditica* has developed a series of mechanisms and strategies to avoid physiological damage, such as the accumulation of phenols, an increase in mycosporine-like amino acids (MAAs), and burial in bottom sediments [2,10]. The photoprotection strategies of cnidarian symbionts against solar radiation play a fundamental role in the efficient recycling of organic and inorganic nutrients between the algae and the host [11]. Endosymbiotic zooxanthellae, through their photosynthetic activity, can provide up to 95% of the daily carbon needs of their hosts. However, under stressful conditions, this supply can be drastically reduced [12,13]. Thus, the efficiency of photosynthetic processes has a direct impact on the functionality of cnidarian–algal symbiosis [14]. However, high levels of light can cause photooxidation and increase the accumulation of ROS, leading to a decrease in photosynthetic activity and ultimately affecting carbon assimilation by endosymbionts [15,16]. Experimental increases in H_2_O_2_ in *A. elegantissima* decreased the Photosystem II (PSII) efficiency of its symbionts [17]. Therefore, understanding the mechanisms of acclimatization to environmental stress in this organism is crucial for understanding its adaptability to different ecosystems.

During the daily light cycle, *A. hermaphroditica* can be exposed to prolonged levels of PAR (400–700 nm) and UVR (280–400 nm), generating an important reduction in the levels of its photochemical parameters owing to the destruction of the D1 protein [18,19]. Studies in the symbiont corals *Acropora pruinosa* and *Pocillopora damicornis* have shown that at high temperatures and radiation levels, there is a reduction in gene expression levels related to the decrease in the photosynthetic efficiency of PSII (F_v_/F_m_) [20]. This increase in ROS by symbiont algae, exacerbated under high-temperature or irradiance conditions, can ultimately lead to bleaching due to the rapid loss of symbionts and/or symbiont pigments [21,22,23]. In contrast, during the night, gene expression levels associated with cellular repair and maintenance processes increase, minimizing the levels of ROS and allowing increases in F_v_/F_m_ parameters [20]. Thus, photosynthetic activity, and therefore the functioning of the cnidarian–algal symbiosis of algal symbionts, is intimately linked to light availability [14]. In *A. hermaphroditica*, under conditions of high temperatures and solar radiation during the summer, increased oxidative damage has been observed, which intensifies under extreme changes in salinity and can ultimately affect photosynthetic performance [2,15,24]. For example, studies on *Stylophora pistillata* and *Galaxea fascicularis* have shown that reduced salinity levels (15-20 PSU) generate a decrease in F_v_/F_m_ levels between 50% and 75% compared with control organisms [25,26]. Understanding the mechanisms underlying tolerance to light and salinity stress in the symbiont *A. hermaphroditica* during daily light and tidal cycles is key to understanding how photoprotective processes enable the success of the symbiotic cnidarian–algal ecological niche in these extreme environments.

In spring-summer, when low tide episodes coincide with the maximum radiation peak in the Quempillén Estuary (13:00–14:00), *A. hermaphroditica* may suffer increased levels of oxidative damage [2,10,24], with a potential effect on photosynthetic performance. Determining the cellular and photochemical responses of this estuarine anemone to daily stress will increase our knowledge of redox metabolism and its influence on the processes of adaptation to environmental stress. The overall objective of this study was to investigate how tidal cycles associated with salinity fluctuations and levels of exposure to ambient radiation in the Quempillén River Estuary influence the photosynthetic and cellular responses of *A. hermaphroditica* and the dinoflagellate symbiont *P. anthopleurum*. Therefore, a field survey was conducted over a 24 h period to determine the photochemical response (F_v_/F_m_, electronic transport rate, photochemical efficiency, and photosynthetic saturation irradiance) and cellular response (oxidative damage/total antioxidant capacity). In addition, two controlled salinity and light experiments were conducted to determine the individual effects of each stress on photooxidative and photochemical responses. This study provides valuable information for understanding how anthozoans adapt to changing environmental conditions, and offers new perspectives for the management and conservation of anthozoans in estuarine areas.

## 2. Materials and Methods

### 2.1. Sample Collection and Acclimation

Adult specimens of *A. hermaphroditica* were collected from the intertidal zone of the Quempillén Estuary (Chiloé Island/41°52′ S; 73°46′ W), during the austral spring-summer (December 2022) Subsequently, they were acclimatized for 24 h to 40 µmol photons m^−2^ s^−1^ (cold daylight; Phillips, Eindhoven, The Netherlands), 12 °C ± 1 °C (water temperature), and a salinity of 33 ± 1 PSU.

### 2.2. Environmental Parameters

Salinity levels in the Quempillén Estuary were measured continuously using a YSI-30 multiparameter. PAR and UVR levels were recorded during the study period using a portable radiometer (Solarlight PM-2000, Glenside, PA, USA) placed over the dock where the outdoor experiments were performed.

### 2.3. Outdoor Experiment

The collected individuals were randomly deposited into food-safe open-top plastic baskets of 80 cm × 60 cm × 30 cm, placed ~5 cm below the surface of the estuarine water in a floating system (Figure 1A), allowing the boxes to remain under the same water recirculation conditions as the estuary. Individuals were exposed to differential ambient radiation using Clear226-UV (Chris James Lighting Fillters, London, UK) and Cellulose di-acetate (CDA; 0.13 mm; Grafix^®^, Maple Heights, OH, USA) cut-off filters for the PAR only and PAR+UVR (280–700 nm) treatments, respectively. The photosynthetic activity of *A. hermaphroditica* was measured every 3 h throughout the experimental period (05:00 a.m. and 02:00 a.m.). The collected samples were preserved in liquid nitrogen and stored at −80 °C before biochemical analysis (antioxidant activity, lipid peroxidation, and phenolic compounds).

### 2.4. Laboratory Experiments

To evaluate the individual effects of changes in salinity and UVR on photochemical responses, collected and previously acclimatized animals were subjected to the following experiments.

#### 2.4.1. Experiment 1: Photosynthesis and UVR

Adults of *A. hermaphroditica* anemone were placed in 24-well microplates (Figure 1B) filled with approximately 10 mL of seawater (33 PSU). Subsequently, the plates were exposed to a set of fluorescent tubes to generate PAR treatment (Daylight Philips, Eindhoven, The Netherlands) and PAR+UVA+UVB treatment (PhilipsTL40 W and Phillips TL20, Philips, Eindhoven, The Netherlands) and covered with Clear226-UV and Cellullose di-acetate cut-off filters. The experimental radiation was set at 2.3 W m^−2^ for UV-B, 8.4 W m^−2^ for UV-A, and 1000 µmol photons m^−2^ s^−1^ for PAR. The animals were exposed to both radiation conditions for 3 h, a period during which high radiation and low tide coincided. After the exposure, the individuals were kept in the dark for 3 h for recovery.

#### 2.4.2. Experiment 2: Photosynthesis and Salinity

Adult anemones were deposited in 24-well microplate wells (Figure 1C), which were filled with approximately 10 mL of seawater at salinities of 30, 22, 15, and 7 PSU. Different salinities were prepared using seawater (33 PSU) diluted with fresh water, and finally corroborated using a portable salinometer YSI-10. The photophysiological parameters (F_v_/F_m_, ETR_max_, α, and E_k_) were determined at the initial (0 h), 24 h, and 48 h.

### 2.5. Measurements of Photosynthetic Activity

The polyphasic increase in Chlorophyll a (Chl-a) fluorescence of each individual *A. hermaphroditica* was measured using the portable PAM fluorometer AquaPen-C AP-C 100 fluorometer (Photon System Instruments, Drásov, Czech Republic). Dark-adapted (15–20 min) samples were placed in a 3 mL fluorescence cuvette that was mounted in front of the detector, while blue excitation light (455 nm) supplied saturating light. Chlorophyll transient light curves were assessed using the OJIP test pre-programmed protocol and were recorded in the time range between 50 μs and 2 s from the onset of light saturation. The OJIP test and calculated parameters were based on those reported by Strasser et al. [27]. Changes in the shape of the curve indicate changes in the function of a specific component of the photosynthetic machinery. The maximal PSII quantum yield was evaluated using the parameter F_v_/F_m_, calculated as F_v_/F_m_ = (F_m_ − F_o_)/F_m_, where F_v_ is the variable fluorescence (F_m_ − F_o_).

The photosynthetic characteristics were assessed by the electron transport rate (ETR) based on photosynthesis vs. irradiance curves (P–I curves), where Φ*_PSII_* was measured as a function of the intensity of actinic irradiance (red-light diode) according to the principles described by Jakob et al. [28]:$$\mathrm{ETR}=\Phi_{PSII}\times E\;\times F_{II}$$
where Φ*_PSII_* is the effective PSII-quantum yield, E is the intensity of the actinic light, and F*_II_* is the fraction of quanta absorbed by PSII (i.e., 0.5). The factor of 0.5 comes from the assumption that four of the eight electrons are required to assimilate one CO_2_ molecule supplied by PSII [29]. A modified nonlinear function [30] was fitted to obtain ETR_max_ (maximal ETR), α (initial slope of the P–I curve as an indicator of photosynthetic efficiency), and E_k_ (saturating irradiance of photosynthesis).

### 2.6. Quantification of Total Phenolic Compounds

Total phenols were determined using the Folin–Ciocalteu method [31], following the protocol modified by Cruces et al. [32]. Approximately 75 mg of tissue previously ground with liquid nitrogen was weighed. The powder was mixed with 1.5 mL 70% (*v*/*v*) aqueous acetone and incubated in an ultrasonic bath for 2 h. Finally, the absorbance was measured at 730 nm using a SPECTROstar Omega microplate reader (BMG LabTech, Ortenberg, Germany), and the results were expressed as milligrams of gallic acid per gram of fresh weight (mg GE g^−1^ FW).

### 2.7. Cellular Responses: Lipid Peroxidation and Antioxidant Capacity

Lipid peroxidation levels were assessed by estimating the malondialdehyde (MDA) concentration in the cell membranes of the *A. hermaphroditica* symbiotic complex according to the protocol of Salama and Pearce [33], with modifications to the microplate [2]. Approximately 30 mg of pulverized frozen tissue was weighed, homogenized with 500 μL of trichloroacetic acid (TCA) (0.1% *w*/*v*), and centrifuged at 4 °C for 10 min at 13,000 RPM. The supernatant was incubated with a mixture of TCA (20% *w*/*v*) and thiobarbituric acid (0.5% *w*/*v*) for 30 min at 80 °C using a thermomixer (Eppendorf, Hauppauge, NY, USA). The final mixture was placed on ice for 5 min and the absorbance was quantified at 522 nm using a SPECTROstar Omega microplate reader (BMG LabTech, BMG, Germany). Lipid peroxidation levels were expressed as nmol MDA g FW^−1^.

The antioxidant activity of the extracts was determined using the 2,2-diphenyl-1-picrylhydrazyl (DPPH) free radical scavenging method [32,34]. DPPH (200 μL volume, 150 μM) was added to each well of a 96-well microplate, 25 μL of the extract of each sample was added, and the absorbance was determined at 532 nm through a kinetic loop for 2 h (37 °C) using a SPECTROstar Omega microplate reader (BMG LabTech, BMG, Germany). Finally, the total antioxidant capacity was estimated using Trolox as a reference and expressed as mg Trolox Eq g FW^−1^.

### 2.8. Statistical Analysis

Data were compared using one-way analysis of variance (ANOVA) for light and salinity treatments, followed by Tukey’s HSD post hoc analysis when differences were detected. Proportions and percentage data were arcsine-transformed to meet ANOVA requirements. ANOVA assumptions (homogeneity of variances and normal distribution) were examined using Levene’s and Shapiro–Wilk W-tests, respectively. Pearson’s test was performed to determine the correlation between variables. Statistical analyses were performed with Origin Pro 2021 version 9.8.0.200 (OriginLab Corporation, Northampton, MA, USA). Statistical significance was set at *p* < 0.05.

## 3. Results and Discussion

### 3.1. Environmental Parameters

During the study period, the salinity levels in the water column fluctuated in relation to tidal changes in the Quempillén Estuary, with salinity reductions of up to 28 PSU at 14:30 and 00:00 (Figure 2A). In contrast, as a result of high tide, values of 33 PSU were reached, which occurred around 6:00 and 18:00 h. A previous study indicated that salt fluctuations resulting from tidal changes in the Quempillén River Estuary during the summer season do not play a fundamental role in cellular and physiological damage to *A. hermaphroditica* [2]. This is supported by the fact that during the austral spring-summer, rainfall decreases considerably [1,2]; therefore, the salinity differences between the high- and low-tide periods are minimal.

Ambient radiation levels during field measurements in the Quempillén River estuary varied in relation to the solar zenith angle, with maximum levels of UV-B, UV-A, and PAR radiation occurring at 13:30 local Chilean (41°52′ S; 73°46′ W) (Figure 2B). The daily maximum irradiance levels for UV-B, UV-A, and PAR were 2.36 W m^−2^, 60.38 W m^2^, and 2025 µmol photons m^−2^ s^−1^, respectively, which is consistent with observations for the same area made by Cubillos et al. [2]. The high ambient radiation levels at midday are determined by the latitudinal location of this estuary in the southern hemisphere, which can receive up to 50% more radiation than its northern hemisphere counterpart, particularly during spring [35]. The diurnal period lasted approximately 14 h, with sunrise at 6:15 am and sunset at 8:15 pm. In addition to the tidal cycles, the effect of solar irradiation on the organisms inhabiting the estuary depends on the bio-optical characteristics of the water column associated with river runoff, seston concentrations, and water exchange processes, which will ultimately determine the depth of action of different wavelengths.(UV-A ~ 4–27 m, UV-B ~ 2–11 m) [36].

### 3.2. Photoacclimation as a Photoprotection Mechanism against Radiation Changes

Throughout the field study (24 h) in the Quempillén River estuary, the photosynthetic parameters of the symbiont complex fluctuated as a function of the daily cycles of PAR and UVR. Thus, the basal values of F_v_/F_m_, ETR_max_, and E_k_ decreased with increasing ambient radiation levels, with the lowest values observed during the hour close to the maximum solar zenith near 14:00 (Figure 3). The decrease in solar radiation levels at the end of the day allowed for the recovery of all photosynthetic parameters exposed only to PAR, whereas PAR+UVR exposure showed, in general, incomplete recovery. The decrease in PSII maximum quantum yield (F_v_/F_m_) resulting from photoinhibitory damage to PSII reaction centers reached a significant decrease of 25% (*p* = 0.021) and 11% (*p* = 0.038) in PAR and PAR+UVR, respectively, compared to the initial values and did not fully recover in PAR+UVR treatments after 6 h of darkness (Figure 3A). Photosynthetic parameters associated with productivity (ETR_max_ and E_k_) were particularly affected by exposure of the symbiont complex to UVR (Figure 3B,C). ETR_max_ and E_k_ during the UVR peak (14:00) decreased by approximately 47% with respect to the initial values (*p* = 0.018 and *p* = 0.009, respectively) but did not reach basal values during the nocturnal recovery period. In contrast, symbiont complexes exposed only to PAR showed an average reduction in ETRmax and E_k_ levels of 35% and 44%, respectively, during the hours of maximum light intensity, which returned to basal levels at night (Figure 3C). Finally, photosynthetic efficiency (α, Figure 3D) fluctuated throughout the study; however, there were no clear effects associated with the different radiation treatments.

The results of the controlled experiments under laboratory conditions showed that PSII maximum quantum yield (F_v_/F_m_) and photosynthetic efficiency (α) were affected by UVR exposure (PAR+UVR, *p* < 0.05) and did not recover at the end of the experiment (Figure 4A,D). However, the parameters associated with productivity were not significantly affected (Figure 4B,D). According to what was observed, the intensity of inhibition in the different photosynthetic parameters is low compared to the results of the field survey (Figure 3), despite the fact that the UVR doses used were close to the environmental doses. Therefore, there are important effects due to exposure time, accumulation of the biologically effective dose (280–400 nm), and radiation intensity that impact the zooxanthella response [37], which is related to the chronic photoinhibition of PSII (F_v_/F_m_) to UVR.

The photoacclimation capacity of *P. anthopleurum* to changes in PAR light intensity was associated with a gradual decrease in photosynthetic parameters (F_v_/F_m_, E_k_, and ETR_max_) during the peak of ambient radiation (14:00) and its subsequent recovery at night. These changes allow it to avoid excess light energy in a harmless manner, thus preventing damage to the photosynthetic apparatus caused by excessive oxidative activity [38]. The photoinhibition capacity (decrease in F_v_/F_m_) is especially important because it downregulates the flux of absorbed photons, maintains an adequate balance between the photoinactivation of PSII and its repair, and allows the recovery of parameters associated with carbon fixation and productivity (ETR_max_ and E_k_) to basal levels [39,40]. In contrast, the photodamage caused by UVR to the photosynthetic apparatus of the dinoflagellate symbiont causes a slowdown in D1 protein repair and ribulosa-1,5-bisfosfato carboxilasa/oxigenasa activity, ultimately causing chronic photoinhibition that affects recovery processes during hours of darkness [22,39,41]. According to Kuguru et al. [42], the mechanisms of tolerance to UVR exposure determine the distribution patterns of cnidarians. Therefore, the PAR and/or UVR tolerance responses of symbionts are species-specific and determine their bathymetric distribution ranges [43]. Similarly, the photophysiological response of the dinoflagellate *P. anthopleurum* is complementary to the behavioral defense of the host cnidarian, which, under high light conditions, may contract, bury itself, or accumulate secondary metabolites with photoprotective capacities as a strategy to minimize physiological stress [13,24,44].

### 3.3. Salinity and Photosynthesis

The effect of salinity and its impact on photosynthesis in the symbiont complex were evaluated through a salinity exposure experiment (30 PSU down to 7 PSU) with 0, 24, and 48 h exposure times. Our results indicate that the symbiont complex could successfully acclimate to different salinities, showing increases in all parameters with respect to the initial values (Figure 5). The differences between the lowest and highest salinity observed between tidal changes during the field survey corresponded to 5 PSU; therefore, the symbiont complex remained at a physiological optimum. Previous studies carried out in the estuarine mollusk *Ostrea chilensis* and gastropod *Crepipatella dilatata* [45,46] indicated that salinities <22 PSU can generate complete isolation of the paleal cavity from the external environment as a measure to reduce osmotic damage to soft tissues, especially to the gill epithelium and hatched embryos. Among the strategies widely used by anemones to cope with salt stress is the expression of heat shock proteins (HSPs), both in the host and symbiont [47]. Schroda et al. [48] indicated that HSP70 chloroplastic proteins are involved in the resistance to photodamage or chronic photoinhibition, reducing the photoinactivation of PSII and enhancing its recovery [47]. Additionally, the internal transport of osmolytes by the host maintains homeostasis within the symbiont complex [49]. Thus, stress induced by salinity changes could potentially trigger a cascade of responses that would enhance host–symbiont responses to other stresses such as UVR or temperature.

### 3.4. Daily Light Cycle and Photooxidative Damage

The sensitivity of the symbiont complex to UVR correlated positively with macromolecular peroxidative damage (r = 0.82; *p* < 0.0001), as evidenced by the significant increase in lipid peroxidation levels in the first hours of exposure to PAR+UVR treatments, reaching a maximum level at 17:00 h, which was approximately four times higher than the initial values (*p* < 0.0001) (Figure 6). At night (grey area in the graph), a significant reduction in lipid peroxidation levels was observed in the PAR+UVR treatment group; however, they did not reach full recovery. In contrast, exposure to PAR radiation during the daytime generated a significant increase (*p* = 0.022) in lipid peroxidation levels starting at 11:00 h (1700 µmol photons m^−2^ s^−1^), reaching a maximum peak at 14:00 h when MDA levels increased up to ~3.7 times compared to the initial values. The decrease in lipid peroxidation levels after 17:00 h was associated with a reduction in PAR levels, allowing full recovery of the levels of oxidative damage previously generated. Changes in PAR light dose across the daily cycle were positively correlated with MDA levels (r = 0.61; *p* < 0.0001) in the symbiont complex. Therefore, peroxidative damage associated with the direct effects of light varies depending on the dose and type of radiation. The direct effect of UVR on target molecules in the early hours of the day induces an increase in peroxidative damage that is maintained throughout the experiment. Direct exposure of the symbiont complex to UVR damages membrane lipids, particularly PSII (i.e., D1 protein), which decreases the ability to dissipate light energy in the dinoflagellate *P. anthopleurum*, increasing the concentration of ROS, and concurrently generating greater cell damage [50,51]. In the case of PAR treatment, during the first hours, there is a balance between the energy absorbed and that used through the electron transport chain of photosynthesis, which is maintained until the increase in the light dose induces the photoinhibition of PSII, a decrease in the electron transport chain, and photosynthetic efficiency, thus increasing the production of ROS. These results account for the importance of photosynthetic processes in the generation of peroxidative tissue damage in this symbiont complex owing to the release of ROS from the dinoflagellate to the host [52,53]. Thus, the ability to counteract the increase in ROS levels by adapting the photosynthetic machinery through the optimal utilization of light energy and the deployment of antioxidant metabolism will allow the recovery of the symbiont complex to face the next diurnal cycle in the Quempillén River estuary.

### 3.5. Increase of Phenolic Compounds and Total Antioxidants during a Daily Cycle

The content of total phenolic compounds showed strong variation with respect to PAR and UVR fluctuations during the daily cycle in the Quempillén Estuary (Figure 7). During the first hours of light (08:00 h), animals exposed to both experimental treatments (PAR and PAR+UVR) increased their concentration of phenolic compounds by ~2.5 times their initial values, showing no significant differences between treatments until 17:00 h, when the PAR+UVR treatment presented significant differences of ~21% with respect to the PAR treatment. The decrease or absence of light was not related to a significant decrease in the concentration of phenolic compounds; therefore, none of the experiments returned to the basal values. No significant correlations were found between salinity changes and phenolic compound levels. In general, cnidarians possess low levels of phenolic compounds, which are acquired mainly through symbiotic associations with zooxanthellae [54]. In this regard, many dinoflagellates are characterized by high concentrations of phenolic compounds [55], and the increase in concentration in response to changes in radiation conditions is associated with their photoacclimatization capacity [56,57]. Among the metabolites that dinoflagellates can produce to cope with environmental variations are flavonoids, terpenoids, and MAAs [58], which mainly serve photoprotective functions either by absorption in the UV range and/or by removing ROS. According to Tarrant et al. [47], the mechanisms of photoprotection in this type of estuarine anemone are determined by circadian rhythms in response to solar and UV radiation. Therefore, the increase in phenolic compounds that is associated with the *A. hermaphroditica*/*P. anthopleurum* symbiont complex is determined by the amount (dose) and types of light during the daily cycle of radiation in the spring-summer season in the Quempillén River estuary.

Between 05:00 and 11:00 h, there was no significant increase in antioxidant activity in either of the two treatments, and only after 14:00 h did the PAR+UVR treatment show evidence of an increase in antioxidant activity, which was maintained until 23:00 h (Figure 8). During the night, the total antioxidant capacity in the PAR+UVR treatment group decreased without reaching basal levels. In the PAR treatment group, the antioxidant activity of the symbiont complex was significantly elevated only from 17:00, a level that remained constant until 02:00. Among the known mechanisms that control the increase in ROS in anemones are a series of antiradical batteries, such as carotenoids, xanthophyll pigments, enzymatic antioxidants (e.g., superoxide dismutase and catalase), and non-enzymatic antioxidants (e.g., vitamin E) [9]. For this study, the increase of antioxidant activity in *A. hermaphroditica* was significantly associated with the concentration of total phenolic compounds, presenting correlations of r = 0.68 and r = 0.65 (*p* < 0.0001) with PAR+UVR and PAR treatments, respectively. The potential reduction of oxidative stress mediated by phenolic compounds is due to their ability to donate electrons to neighboring hydroxyl groups that allow them to act as reducing agents, hydrogen, and singlet oxygen donors [59,60]. Recently, De Domenico et al. [61], through a study on the jellyfish *Cassiopea andromeda*, which harbors autotrophic dinoflagellate symbionts of the Symbiodiniaceae family, determined the contribution of these microalgae to bioactive compounds, such as polyphenols and pigments, with antioxidant properties and absorption ranges below 400 nm. Although the compounds were not identified, the absorption spectra showed the UV absorbance ranges of the phenolic subfamilies, that is, phenolic acids (270–280 nm and 305–330 nm) and flavonoids (270–280 and 310–350 nm). This study also revealed that the highest concentrations of phenolic compounds were located in the tentacles of anemones, where the highest photosynthetic activity was associated with the endosymbiotic zooxanthellae [61]. In symbiotic dinoflagellates and cnidarians, the most recognized and studied anti-stress compounds are MAAs, which are secondary metabolites that not only have photoprotective capacities and are able to absorb wavelengths between 295 and 365 nm (UV-B, UV-A), but also have antioxidant capacities [62,63]. In *A. hermaphroditica*, the presence of mycosporin 2-glycine (Myc 2-gly), a common compound among symbiotic anemones that can absorb wavelengths in the UVA range because of its λmax of 332 nm, has been identified [2,64]. The presence of MAAs in symbiotic invertebrates is mainly associated with their diet; however, this would not explain their potential increase under conditions of high light stress. In this regard, several studies on symbiotic dinoflagellates have reported adjustments in the amount of MAAs synthesized and accumulated in response to light stress, specifically UV radiation [65,66,67,68]. Therefore, it is presumed that the main contributing factors that increase in this compound are determined by the symbiotic zooxanthellae *P. anthopleurum* [69].

In conclusion, it was found that the wide latitudinal distribution of the *A. hermaphroditica*/*P. anthopleurum* symbiont complex along the Chilean coast may be due to its high acclimation capacity and the development of efficient defensive systems in response to local environmental conditions. Light is an essential regulatory factor affecting the productivity, physiology, and ecology of this symbiont anemone as well as influencing its nutritional processes, as an important part of its diet comes from photosynthates generated by its symbiotic zooxanthellae [70]. Therefore, determining the intensity and duration of light required to trigger stress reactions at the photosynthetic apparatus level, as well as the impact of light/dark alternation on damage and repair processes, is crucial for understanding the mechanisms that allow these organisms to survive and thrive in the dynamic estuarine ecosystems of southern Chile. In this study, a series of metabolic adjustments between the zooxanthella and anemone minimized the harmful effects of UVR, a modulator of anti-stress responses in this organism. The cellular and physiological capacities to counteract potential oxidative damage are associated with the adjustment of photosynthetic metabolism, mitigation mechanisms, and antioxidant defenses during light stress. In particular, stress tolerance under increasing light levels towards midday was mediated by dynamic photoinhibition and then complemented by increased phenolic compounds and an increased antioxidant system in the second part of the day. However, in the presence of high UVR, there was an increase in damage that slowed down the reparative processes during the night. The cellular/physiological plasticity of the symbiont complex is complemented by behavioral responses to UVR stress, including tentacular crown retraction or burial in the sediment, which are highly effective in reducing photo-oxidative damage [10]. In this regard, this study also raises questions about the variation in responses between organisms adapted to intertidal versus subtidal and juvenile versus adult conditions, and how this species will respond to long-term environmental fluctuations (e.g., those caused by climate change). Finally, according to our results, solar radiation, particularly UVR, is an abiotic factor that drives the physiological cycles of the environmental stress responses in this symbiont complex.

## Figures and Tables

**Figure 1 antioxidants-13-01239-f001:**
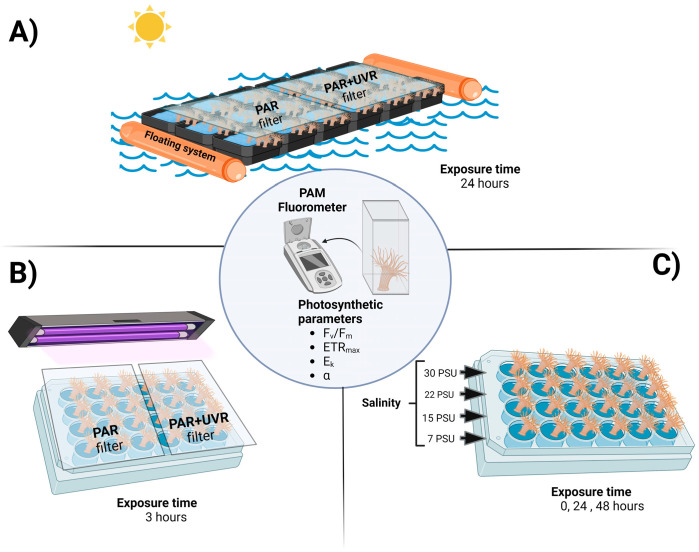
Experimental design for *A. hermaphroditica* exposed in a floating system to a daily cycle of light (PAR and PAR+UVR) and tide in the field during a 24 h period in the Quempillén estuary (**A**); short term exposure to experimental radiation (PAR and PAR+UVR) (**B**); and experimental salinities (**C**) under laboratory conditions. Photosynthetic parameters were assessed in all experimental animals using a portable fluorometer. Figure created using BioRender.com.

**Figure 2 antioxidants-13-01239-f002:**
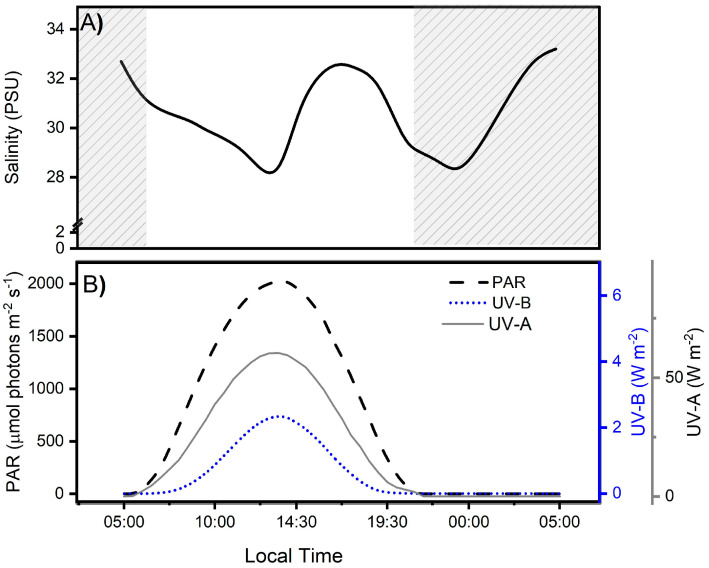
Salinity levels (**A**) and solar irradiance during the experimental period (**B**), separated for photosynthetically active (PAR: 400–700 nm), ultraviolet-B (UV-B: 280–315 nm), and ultraviolet-A (UV-A: 315–400 nm) radiation (**B**) in the Quempillén River estuary (41°52′ S; 73°46′ W) during a 24 h period for a clear spring-summer day (December 2022). The gray area corresponds to hours of the day without light.

**Figure 3 antioxidants-13-01239-f003:**
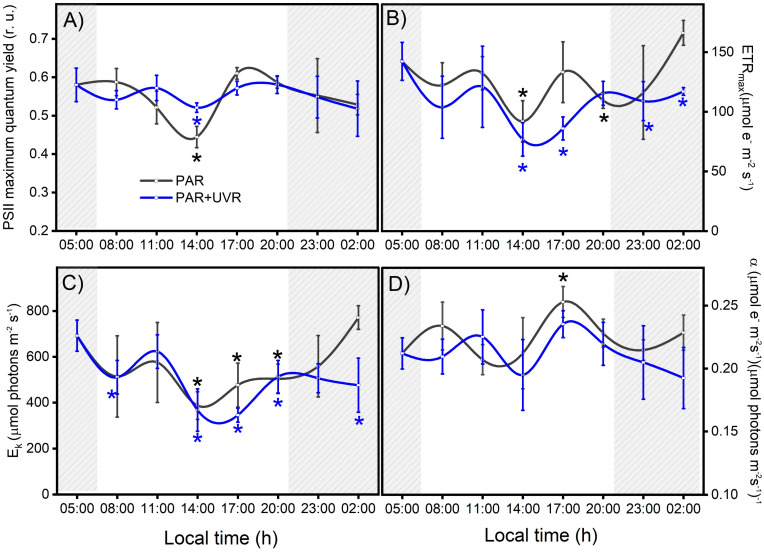
Variation in photosynthetic activity in the symbiotic complex *A. hermaphroditica*/*P. anthopleurum* exposed to PAR (gray line) and PAR+UVR (blue line) radiation treatments. The parameters correspond to PSII maximal quantum yield for electron transport (**A**), the maximal photosynthetic electron transport rate (**B**), the photosynthetic efficiency (**C**), and the light saturation index (**D**) during a daily light–tide cycle. The gray area corresponds to hours of the day without light. These Chl-a fluorescence parameters were calculated from the P–I light saturation curve. Data are presented as mean ± standard deviation (*n* = 3). Asterisks above or below the data points indicate a statistical difference compared to baseline *p* ˂ 0.05), in black or blue for PAR or PAR+UVR, respectively.

**Figure 4 antioxidants-13-01239-f004:**
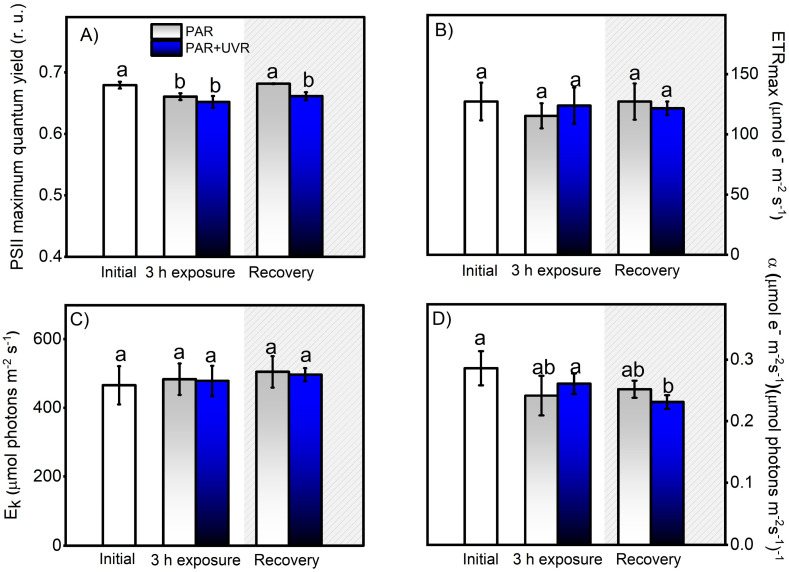
Change in photosynthetic activity, indicated by the PSII maximal quantum yield for electron transport (**A**), the maximal photosynthetic electron transport rate (**B**), the photosynthetic efficiency (**C**), and the light saturation index (**D**). These Chl-a fluorescence parameters were calculated from the P–I light saturation curve in control *A. hermaphroditica* adults and in individuals exposed for 3 h to PAR (blue bars) and PAR+UVR (gray bars) conditions and then recovery in darkness for 3 h (coarse area). Data are presented as mean ± standard deviation (*n* = 3). Letters above bars represent statistically significant differences between groups as determined by ANOVA and Tukey’s HSD tests (*p* ˂ 0.05).

**Figure 5 antioxidants-13-01239-f005:**
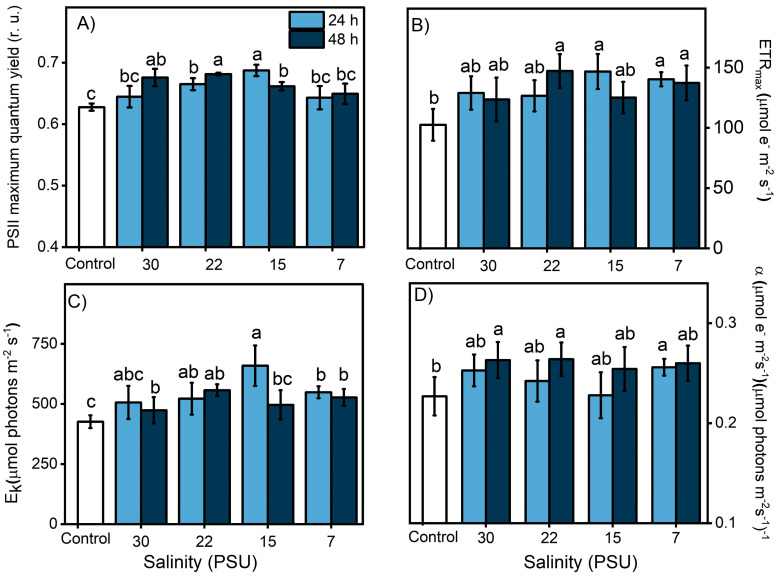
Change in photosynthetic activity of the symbiotic complex *A. hermaphroditica/P. anthopleurum*, indicated by the PSII maximal quantum yield for electron transport (**A**), the maximal photosynthetic electron transport rate (**B**), the photosynthetic efficiency (**C**), and the light saturation index (**D**). These Chl-a fluorescence parameters were calculated from the P–I light saturation curve in control organisms exposed during 24 h (light blue bars) and 48 h (dark blue bars) to experimental salinities (30, 22, 15 and 7 PSU). Data are presented as mean ± standard deviation (*n* = 3). Letters above bars represent statistically significant differences between groups as determined by ANOVA and Tukey’s HSD tests (*p* < 0.05).

**Figure 6 antioxidants-13-01239-f006:**
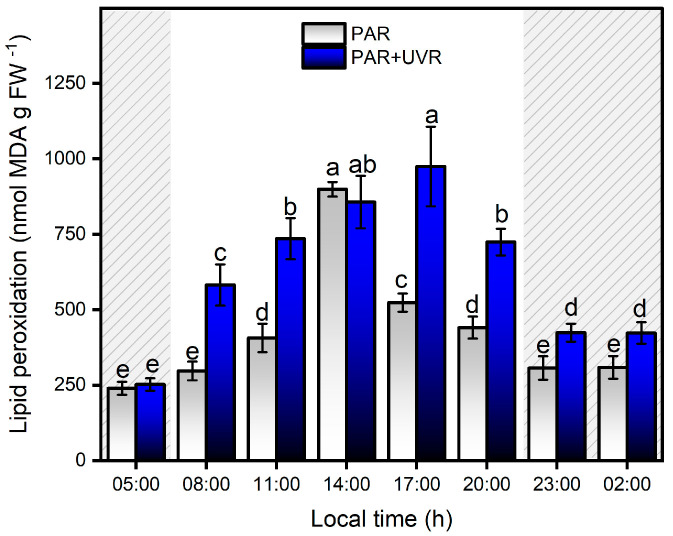
Levels of lipid peroxidation levels in the symbiotic complex *A. hermaphroditica/P. anthopleurum* exposed to PAR (blue bars) and PAR+UVR (gray bars) during a daily light/tidal cycle in the Quempillén estuary throughout a 24 h period. Coarse area corresponds to hours of the day without light (night period). Data are presented as mean ± standard deviation (*n* = 6). Letters above bars represent statistically significant differences between groups as determined by ANOVA and Tukey’s HSD tests (*p* < 0.05).

**Figure 7 antioxidants-13-01239-f007:**
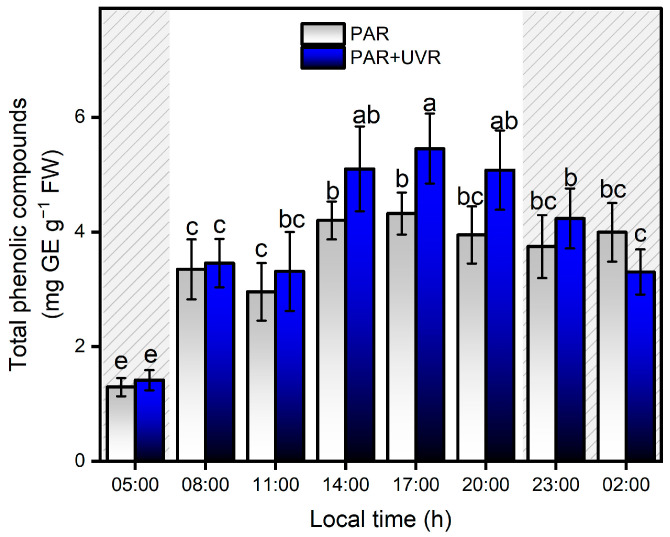
Levels of total phenolic compounds in the symbiotic complex *A. hermaphroditica/P. anthopleurum* exposed to PAR (blue bars) and PAR+UVR (gray bars) during a daily light/tidal cycle in the Quempillén estuary throughout a 24 h period. Coarse area corresponds to hours of the day without light (night period). Data are presented as mean ± standard deviation (*n* = 6). Letters above bars represent statistically significant differences between groups as determined by ANOVA and Tukey’s HSD tests (*p* < 0.05).

**Figure 8 antioxidants-13-01239-f008:**
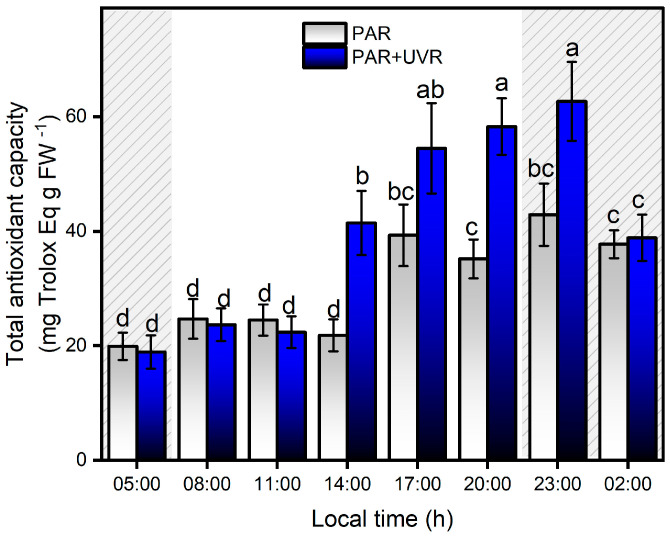
Total antioxidant capacity levels in the symbiotic complex *A. hermaphroditica/P. anthopleurum* exposed to PAR (blue bars) and PAR+UVR (gray bars) during a daily light/tidal cycle in the Quempillén estuary through a 24 h period. Coarse area corresponds to hours of the day without light (night period). Data are presented as mean ± standard deviation (*n* = 6). Letters above bars represent statistically significant differences between groups as determined by ANOVA and Tukey’s HSD tests (*p* < 0.05).

## Data Availability

The data presented in this study are available in a data repository.

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
