# Peer review of "Photophysiological and Oxidative Responses of the Symbiotic Estuarine Anemone Anthopleura hermaphroditica to the Impact of UV Radiation and Salinity: Field and Laboratory Approaches"

_antioxidants, 2024, doi:10.3390/antiox13101239_

Round 1

Reviewer 1 Report

Comments to the Author:

The authors researched the responding mechanism of the symbiotic estuarine anemone Anthopleura hermaphroditica, in face of the stresses of UV radiation and salinity. The results will provide a guidance for ecological and conservation efforts of estuaries. However, the paper is not innovative enough, some important conclusions have been reported. The research method used is relatively conventional, which can not deeply analyze the adaptation mechanism of the symbiont complex, responding to changes in various environment conditions. Therefore, some suggestions and revisions are needed for the paper before it could be preferably accepted.

1. In line 477-479, the authors indicated that solar radiation, particularly UVR, is an abiotic factor that drives the physiological cycles of the environmental stress responses in this symbiont complex. However, the related conclusion had been described in line 83-86, In A. hermaphroditica, under conditions of high temperatures and solar radiation during the summer, increased oxidative damage has been observed, which intensifies under extreme changes in salinity and can ultimately affect photosynthetic performance [17,26,27]. So I propose to further summarize the innovation points of this paper.

2. In line 340-342, “Our results indicate that the symbiont complex could successfully acclimate to different salinities, showing increases in all parameters with respect to the initial values (Fig. 5).”. The authors performed a salinity exposure experiment (30 PSU to7 PSU), and found that this salinity range had no significant effect on the symbiont complex. So I suggest adding salinity tolerance tests to determine the salinity tolerance range of the symbiont complex.

3. In line 450-452, “Therefore, it is presumed that the main contribution to increase in this compound are determined by the symbiotic zooxanthellae P. anthopleurum [68].. This hypothesis is so interesting. If relevant supporting evidence can be found, the quality of this paper will be improved.

4. In line 442-445, and line 463-464. The vital role of metabolites produced by the symbiont complex was highlighted by numerous researches of photophysiological and oxidative responses. In this paper, the regulatory mechanism of the symbiont complex, responding to UV radiation and salinity stresses, was only explored by total phenolic compounds, lipid peroxidation and antioxidant capacity. So I suggest adding metabolite detection and stress-responsive gene expression experiments.

Author Response

The authors researched the responding mechanism of the symbiotic estuarine anemone Anthopleura hermaphroditica, in face of the stresses of UV radiation and salinity. The results will provide a guidance for ecological and conservation efforts of estuaries. However, the paper is not innovative enough, some important conclusions have been reported. The research method used is relatively conventional, which cannot deeply analyze the adaptation mechanism of the symbiont complex, responding to changes in various environment conditions. Therefore, some suggestions and revisions are needed for the paper before it could be preferably accepted.

Detail comments

  1. In line 477-479, the authors indicated that solar radiation, particularly UVR, is an abiotic factor that drives the physiological cycles of the environmental stress responses in this symbiont complex. However, the related conclusion had been described in line 83-86, “In  hermaphroditica, under conditions of high temperatures and solar radiation during the summer, increased oxidative damage has been observed, which intensifies under extreme changes in salinity and can ultimately affect photosynthetic performance [17,26,27]. So I propose to further summarize the innovation points of this paper.

Responses: We thank the reviewer for their comment. In this regard, we have included some points regarding the importance of light as a stress-regulating factor for the A. hermaphroditica/P Anthopopleurum symbiont complex under natural conditions. We have further emphasized the novelty of this work in the final section of the revised manuscript. See lines 480-491:

In conclusion, it was found that the wide latitudinal distribution of the A. hermaphroditica/P. anthopleurum symbiont complex along the Chilean coast may be due to its high acclimation capacity and the development of efficient defensive systems in response to local environmental conditions. Light is an essential regulatory factor affecting the productivity, physiology, and ecology of this symbiont anemone as well as influencing its nutritional processes, as an important part of its diet comes from photo-synthates generated by its symbiotic zooxanthellae [70]. Therefore, determining the intensity and duration of light required to trigger stress reactions at the photosynthetic apparatus level, as well as the impact of light/dark alternation on damage and repair processes, is crucial for understanding the mechanisms that allow these organisms to survive and thrive in the dynamic estuarine ecosystems of southern Chile. In this sense, the series of metabolic adjustments of zooxanthella and anemone minimized the harmful effects of UVR, a modulator of anti-stress responses in this organism.

  1. In line 340-342, “Our results indicate that the symbiont complex could successfully acclimate to different salinities, showing increases in all parameters with respect to the initial values (Fig. 5).”. The authors performed a salinity exposure experiment (30 PSU to7 PSU), and found that this salinity range had no significant effect on the symbiont complex. So I suggest adding salinity tolerance tests to determine the salinity tolerance range of the symbiont complex.

Response: The authors sincerely appreciate the reviewer's valuable comments to improve the quality of the manuscript. In this regard, studies on the A. hermaphroditica symbiont complex conducted by our group (Cubillos et al. 2018) considered the seasonal responses of this organism in relation to UVR, temperature, and salinity. This study found that during summer, the main factors that determine stress are temperature and UVR, unlike in winter, in which salinity is the factor that determines the highest stress levels in the symbiont anemone. Therefore, the main objective of our study was to evaluate the effects of radiation and salinity changes on the photosynthetic processes of zooxanthellae in animals acclimated to early summer conditions, and how this stress is related to the photophysiological responses of the symbiont complex.  In general, the effect of salinity during this period was minor, considering that the lowest estuarine salinities were well above the limits of salinity tolerance observed in this species (<22 PSU). 

  1. In line 450-452, “Therefore, it is presumed that the main contribution to increase in this compound are determined by the symbiotic zooxanthellae  anthopleurum[68].”. This hypothesis is so interesting. If relevant supporting evidence can be found, the quality of this paper will be improved.

Response : We sincerely appreciate the reviewer's valuable comments on improving the quality of the manuscript. In this regard, there are few studies on the contribution of phenolic compounds from the zooxanthellae to the photoprotection of the symbiont complex.  De Domenico et al (2023) through a study in the holobiont zooxanthellate jellyfish Cassiopea andromeda that harbors autotrophic dinoflagellate symbionts (family Symbiodiniaceae) determined the contribution of these microalgae with respect to bioactive compounds such as polyphenols and pigments, including carotenoids, with antioxidant properties and with absorption ranges below 400 nm. This study also showed that the highest concentrations of phenolic compounds were associated with endosymbiotic zooxanthellae. Regarding the potential increase in MAAs by symbiont dinoflagellates due to light stress, several studies have reported these adjustments in this type of organism, especially under different doses of UV radiation (Jefreey 1999; Rastogi et al. 2010; Shick 2004; Hoshihara et al. 2022). We rewrote the text incorporating the reviewer's suggestions. The text now reads (450-459 and 467-469):

Recently, De Domenico et al. [61], through a study on the jellyfish Cassiopea andromeda, which harbors autotrophic dinoflagellate symbionts of the Symbiodiniaceae family, determined the contribution of these microalgae to bioactive compounds, such as polyphenols and pigments, with antioxidant properties and absorption ranges below 400 nm. Although the compounds were not identified, the absorption spectra showed the UV absorbance ranges of the phenolic subfamilies, that is, phenolic acids (270-280 nm and 305-330 nm) and flavonoids (270-280 and 310-350 nm). This study also revealed that the highest concentrations of phenolic compounds were located in the tentacles of anemones, where the highest photosynthetic activity was associated with the endo-symbiotic zooxanthellae [61].

In this regard, several studies on symbiotic dinoflagellates have reported adjustments in the amount of MAAs synthesized and accumulated in response to light stress, specifically UV radiation [65–68].

  1. In line 442-445, and line 463-464. The vital role of metabolites produced by the symbiont complex was highlighted by numerous researches of photophysiological and oxidative responses. In this paper, the regulatory mechanism of the symbiont complex, responding to UV radiation and salinity stresses, was only explored by total phenolic compounds, lipid peroxidation and antioxidant capacity. So I suggest adding metabolite detection and stress-responsive gene expression experiments.

Response: The authors sincerely appreciate the reviewer's valuable comments to improve the quality of the manuscript. However, the main objective of this study was to determine the photophysiological (Fv/Fm, ETRmax, Ek, and α of PSII) and cellular (oxidative damage and antioxidant response) responses of the symbiont complex formed by the anemone Anthopleura hermaphroditica and the dinoflagellate Philozoon anthopleurum to solar radiation and salinity stress during a 24-hour day/night cycle. Undoubtedly, the detection of metabolites and stress-responsive gene expression experiments are very interesting for understanding the underlying mechanisms associated with these organisms, and we hope to address these in future studies.

Reviewer 2 Report

This manuscript investigated the photophysiological and oxidative responses of the symbiotic   estuarine anemone Anthopleura hermaphroditica to the impact of UV radiation and salinity. They found the photosynthetic parameters of the symbiont complex decreased with increasing ambient radiation; however, no relationship was observed with changes in salinity. During the dark period, only PAR-exposed organisms returned to basal levels of photosynthesis and cell damage. The ability of A to acclimate to natural radiation stress is mediated by the concerted action of various physiological mechanisms that occur at different times of the day under varying levels of environmental stress. This is an informative and interesting manuscript.

1. The abstract is a brief introduction to the entire article, but it is poorly written. For example, the sentence ' Additionally, photobio-logical responses to photosynthetically active radiation (PAR: 400–700 nm), PAR + ultraviolet radi-ation (UVR: 280–400 nm), and salinity were determined through laboratory experiments' is confusing, the sentence' The increase in peroxidative damage, total phenolic compounds, and antioxidant activity were functions of the level and type of environmental radiation' needs to explain what function it is, and the abstract lacks a convincing summary.

2. Abbreviations should be defined or written in full when they first appear. Such as "PSII" in line 68, "REDOX" in Line 97..., please double check all the text to find similar errors and correct the5.

3. Please unify the format of references in the article, including the author's name, the case of words in the title of the article, the writing of the name of the journal, and the page number.

4. Figure 6, Why is the trend of lipid peroxidation levels at 12:00 different from other time points.

5. Figure 8 is missed.

Author Response

This manuscript investigated the photophysiological and oxidative responses of the symbiotic   estuarine anemone Anthopleura hermaphroditica to the impact of UV radiation and salinity. They found the photosynthetic parameters of the symbiont complex decreased with increasing ambient radiation; however, no relationship was observed with changes in salinity. During the dark period, only PAR-exposed organisms returned to basal levels of photosynthesis and cell damage. The ability of A to acclimate to natural radiation stress is mediated by the concerted action of various physiological mechanisms that occur at different times of the day under varying levels of environmental stress. This is an informative and interesting manuscript.

Response: We thank the reviewer for the positive comments and addressed point by point the different elements raised by the reviewer in the response below.

Detail comments

  1. The abstract is a brief introduction to the entire article, but it is poorly written. For example, the sentence ' Additionally, photobiological responses to photosynthetically active radiation (PAR: 400–700 nm), PAR + ultraviolet radiation (UVR: 280–400 nm), and salinity were determined through laboratory experiments' is confusing, the sentence' The increase in peroxidative damage, total phenolic compounds, and antioxidant activity were functions of the level and type of environmental radiation' needs to explain what function it is, and the abstract lacks a convincing summary.

Response: The authors are thankful for the valuable comment given by the reviewer to enhance the manuscript's quality. In accordance with the reviewer's comments, the abstract was improved and confusing sentences were corrected. The required changes have been incorporated in the revised manuscript and the same has been given below. The text now reads (15-30):

The estuarine anemone Anthopleura hermaphroditica and its symbiont Philozoon anthopleurum are continuously exposed to intense fluctuations in solar radiation and salinity owing to tidal changes. The aim of this study was to evaluate the effects of the tidal cycle, solar radiation, and salinity fluctuations on the photosynthetic and cellular responses (lipid peroxidation, total phenolic compounds, and antioxidant activity) of the symbiont complex over a 24-hour period in the Quempillén River Estuary. Additionally, laboratory experiments were conducted to determine the specific photobiological responses to photosynthetically active radiation (PAR), ultraviolet radiation (UVR), and salinity. Our field results showed that the photosynthetic parameters of the symbiont complex decreased with increasing ambient radiation; however, no relationship was observed with changes in salinity. Increased peroxidative damage, total phenolic compound levels, and antioxidant activity were mainly related to increased UVR and, to a lesser extent, PAR. During the dark period, only PAR-exposed organisms returned to the basal levels of photosynthesis and cell damage. Laboratory exposure confirmed the deleterious effects of UVR on the photosynthetic response. The present study suggests that the ability of A. hermaphroditica to acclimate to natural radiation stress is mediated by the concerted action of various physiological mechanisms that occur at different times of the day, under varying levels of environmental stress.

  1. Abbreviations should be defined or written in full when they first appear. Such as "PSII" in line 68, "REDOX" in Line 97..., please double check all the text to find similar errors and correct the5.

Response: The authors are grateful for the valuable comments provided by the reviewer to improve the quality of the manuscript. In accordance with the reviewer's comments, the definitions of the abbreviations have been incorporated in order of their appearance in the text.

  1. Please unify the format of references in the article, including the author's name, the case of words in the title of the article, the writing of the name of the journal, and the page number.

Response: The authors are grateful for the comments provided and the format of the references was revised and formatted according to the journal's guidelines.

  1. Figure 6, Why is the trend of lipid peroxidation levels at 12:00 different from other time points.

Response: The authors sincerely appreciate the reviewer's detailed and constructive comments, which have helped us recognize areas for improvement and improved the quality of the manuscript. We have provided a more detailed explanation for the differential effects of light on peroxidative damage. According to our results, the peroxidative damage associated with the direct effects of light varies according to the dose and type of radiation.  In the case of PAR, during the first few hours, there is a balance between the energy absorbed and that utilized through the electron transport chain of photosynthesis, which is maintained until the increase in light dose produces photoinhibition of PSII, decreasing electron transport and photosynthetic efficiency, consequently increasing the production of ROS inducing oxidative damage. In the case of UVR, the direct effect of these wavelengths in the early hours of the day on lipid membranes and D1 protein produced an increase in peroxidative damage from 11:00 h onwards, which was maintained throughout the field survey. Details of the explanation have been included in the revised manuscript between lines 380-393, as detailed below.

Therefore, peroxidative damage associated with the direct effects of light varies de-pending on the dose and type of radiation. The direct effect of UVR on target molecules in the early hours of the day induces an increase in peroxidative damage that is maintained throughout the experiment. Direct exposure of the symbiont complex to UVR damages membrane lipids, particularly PSII (i.e., D1 protein), which decreases the ability to dissipate light energy in the dinoflagellate P. anthopleurum, increasing the concentration of ROS, and concurrently generating greater cell damage [50,51]. In the case of PAR treatment, during the first hours, there is a balance between the energy absorbed and that used through the electron transport chain of photosynthesis, which is maintained until the increase in the light dose induces photoinhibition of PSII, a decrease in the electron transport chain, and photosynthetic efficiency, increasing the production of ROS. These results account for the importance of photosynthetic processes in the generation of peroxidative tissue damage in this symbiont complex owing to the release of ROS from the dinoflagellate to the host [52,53].

  1. Figure 8 is missed.

Response : Thank you for your comment, the number in figure 8 has been corrected in the manuscript.

Reviewer 3 Report

This is an interesting manuscript on the tolerance of a sea anemone to tidal changes, including radiation and salinity, assessed by studying photosynthetic parameters and oxidative stress biomarkers. Overall, it is a well written manuscript with important findings in the area of not only sea anemones but in cnidarians in general, including corals. The only potential limitation is the short-time period studied for the field component of this work. Another similar scenario under other tidal regime, moon cycle could be compared.

Lines 37-39. This sentence is unclear, it needs rewording.

Line 39. “52” is missing the minute symbol, and “S” for South should be in capital letter.

Line 48. Please check the spelling of the dinoflagellate species; reference 6 spells the genus as “Philozoon”.

Line 55. Correct the species name.

Line 88. It should read: …salinity levels (15-20 PSU)…

Line 94. It is the oxidative damage itself that increases in the anemone, or the anemone is subject to (or suffers) an oxidative damage increase. But “the anemone does not increase the levels of oxidative damage”. Please rewrite.

Lines 209-216. Please include the software used to carry out the statistical analysis.

Line 236. Given that Chile covers a wide range of latitudes, it would be good to mention again the coordinates after “local Chilean”.

Lines 256, 279, 370, 373 and elsewhere. I recommend including the exact p-value returned by the statistical analysis software. Here and across the manuscript.

Line 339. Generally, it is written from the low to the high value, e.g. 7 to 30 PSU. However, I understand that the authors wanted to simulate the impact of rain or run-off (i.e. salinity dilution effect). In this case, please correct to “(30 PSU down to 7 PSU)

Line 460. I recommend starting the last paragraph as “In conclusion, it was found that the wide latitudinal…

Author Response

This is an interesting manuscript on the tolerance of a sea anemone to tidal changes, including radiation and salinity, assessed by studying photosynthetic parameters and oxidative stress biomarkers. Overall, it is a well written manuscript with important findings in the area of not only sea anemones but in cnidarians in general, including corals. The only potential limitation is the short-time period studied for the field component of this work. Another similar scenario under other tidal regime, moon cycle could be compared.

Response: The authors are grateful for the reviewer's valuable comments. In this regard, studies on the symbiont complex A. hermaphroditica conducted by our study group (Cubillos et al. 2018) considered the seasonal responses of this symbiont complex in relation to UV radiation, temperature, and salinity. The objective of our study was to evaluate how changes in radiation (PAR/UVR) and salinity modulated by tidal changes affect the photophysiological and cellular responses of a symbiotic complex composed of sea anemones and their dinoflagellates. We agree that studies on the seasonal effects of lunar cycles on the intensity of tidal fluctuations are needed. However, environmental conditions in the field, especially in southern Chile, make this type of study very difficult. Thus, the full moon/new moon vs. first/third quarter contrast can be a bit complex because environmental conditions can change (continuous precipitation, clear sky day, foggy conditions, etc.).   

Detail comments

Lines 37-39. This sentence is unclear, it needs rewording.

Response : Thank you for your comment, the sentence has been corrected. The text now reads (36-38):

Similarly, diurnal tidal changes in these ecosystems can modify the availability of light, salinity, and temperature of the water column over a short period of time.

Line 39. “52” is missing the minute symbol, and “S” for South should be in capital letter.

Response : Thanks for the comment the symbol has been changed as suggested

Line 48. Please check the spelling of the dinoflagellate species; reference 6 spells the genus as “Philozoon”.

Response : The authors are grateful for this reviewer's comments and the error in the genus of zooxanthellae has been corrected in the manuscript.

Line 55. Correct the species name.

Response: Thank you for your comment, the species name has been corrected.

Line 88. It should read: …salinity levels (15-20 PSU)

Response: Thank you for your comment, the sentence has been amended.

Line 94. It is the oxidative damage itself that increases in the anemone, or the anemone is subject to (or suffers) an oxidative damage increase. But “the anemone does not increase the levels of oxidative damage”. Please rewrite.

Response : Thanks for the comment, the sentence was corrected in the manuscript.  The text now reads (94-95):

In spring-summer, when low tide episodes coincide with the maximum radiation peak in the Quempillén Estuary (13:00-14:00), A. hermaphroditica may suffer increased levels of oxidative damage

Lines 209-216. Please include the software used to carry out the statistical analysis.

Response : Thank you for your comment. The statistical software was included. The text now reads (216-217):

Statistical analyses were performed with Origin Pro 2021 version 9.8.0.200 (OriginLab Corporation).

Line 236. Given that Chile covers a wide range of latitudes, it would be good to mention again the coordinates after “local Chilean”.

Response: The authors are grateful for the reviewer's suggestion, which was included in the manuscript. The text now reads (238):

Ambient radiation levels during field measurements in the Quempillén River estuary varied in relation to the solar zenith angle, with maximum levels of UV-B, UV-A, and PAR radiation occurring at 13:30 local Chilean (41°52ꞌS; 73°46ꞌ W)

Lines 256, 279, 370, 373 and elsewhere. I recommend including the exact p-value returned by the statistical analysis software. Here and across the manuscript.

Response:The authors are grateful for the reviewer's suggestion. The exact p-values were included in the manuscript, except in those cases where the significance was less than 0.0001 where the symbology p˂ 0.0001 was used.

Line 339. Generally, it is written from the low to the high value, e.g. 7 to 30 PSU. However, I understand that the authors wanted to simulate the impact of rain or run-off (i.e. salinity dilution effect). In this case, please correct to “(30 PSU down to 7 PSU)

Response: Thanks for the comment, the sentence has been corrected in the manuscript as suggested by the reviewer.

Line 460. I recommend starting the last paragraph as “In conclusion, it was found that the wide latitudinal…

Response: Thanks for the comment, the sentence has been corrected in the manuscript as suggested by the reviewer.

References included in the corrected manuscript

[61]S. De Domenico, G. De Rinaldis, M. Mammone, M. Bosch-Belmar, S. Piraino, A. Leone, The Zooxanthellate Jellyfish holobiont Cassiopea andromeda, a source of soluble bioactive compounds, Mar. Drugs. 21 (2023) 1–22. https://doi.org/10.3390/md21050272.

[65]       S.W. Jeffrey, H.S. Mactavish, W.C. Dunlap, M. Vesk, K.M. Groenewoud, Occurrence of UVA-and UVB-absorbing compounds in 152 species (206 strains) of marine microalgae, 189 (1999) 35–51. http://www.int-res.com/articles/meps/189/m189p035.pdf.

[66]     R.P. Rastogi, Richa, R.P. Sinha, S.P. Singh, D.P. Häder, Photoprotective compounds from marine organisms, J. Ind. Microbiol. Biotechnol. 37 (2010) 537–558. https://doi.org/10.1007/s10295-010-0718-5.

[67]     J.M. Shick, The continuity and intensity of ultraviolet irradiation affect the kinetics of biosynthesis, accumulation, and conversion of mycosporine-like amino acids (MAAs) in the coral Stylophora pistillata, Limnol. Oceanogr. 49 (2004) 442–458. https://doi.org/10.4319/lo.2004.49.2.0442.

[68]     A. Yamamoto Hoshihara, T. Fujiki, S. Shigeoka, M. Hirayama, Physiological response of the symbiotic dinoflagellate Pelagodinium béii to ultraviolet radiation: synthesis and accumulation of mycosporine-like amino acids, Symbiosis. 86 (2022)

[70]    J. Helgoe, S.K. Davy, V.M. Weis, M. Rodriguez-Lanetty, Triggers, cascades, and endpoints: connecting  the dots of coral bleaching mechanisms, Biol. Rev. 99 (2024) 715–752. https://doi.org/10.1111/brv.13042.

Round 2

Reviewer 1 Report

The authors researched the responding mechanism of Anthopleura hermaphroditica, in face of the stresses of UV radiation and salinity, which is conductive to ecological protection of estuaries. I suggested that the manuscript could be accepted in present form.

The authors researched the responding mechanism of Anthopleura hermaphroditica, in face of the stresses of UV radiation and salinity, which is conductive to ecological protection of estuaries. I suggested that the manuscript could be accepted in present form.